# Emergence of high piezoelectricity from competing local polar order-disorder in relaxor ferroelectrics

Hui Liu [1,2,9] ✉, Xiaoming Shi[3,9], Yonghao Yao[1,2], Huajie Luo[1,2], Qiang Li[1], Houbing Huang [3] ✉, He Qi [1,2], Yuanpeng Zhang[4], Yang Ren[5], Shelly D. Kelly[6], Krystian Roleder [7], Joerg C. Neuefeind [4], Long-Qing Chen [8], Xianran Xing [1] & Jun Chen [1,2] ✉

Relaxor ferroelectrics are known for outstanding piezoelectric properties, finding a broad range of applications in advanced electromechanical devices. Decoding the origins of the enhanced properties, however, have long been complicated by the heterogeneous local structures. Here, we employ the advanced big-box refinement method by fitting neutron-, X-ray-based total scattering, and X-ray absorption spectrum simultaneously, to extract local atomic polar displacements and construct 3D polar configurations in the classical relaxor ferroelectric $Pb(Mg_{1/3}Nb_{2/3})O_3$–$PbTiO_3$. Our results demonstrate that prevailing order-disorder character accompanied by the continuous rotation of local polar displacements commands the composition-driven global structure evolution. The omnidirectional local polar disordering appears as an indication of macroscopic relaxor characteristics. Combined with phase-field simulations, it demonstrates that the competing local polar order-disorder between different states with balanced local polar length and direction randomness leads to a flattening free-energy profile over a wide polar length, thus giving rise to high piezoelectricity. Our work clarifies that the critical structural feature required for high piezoelectricity is the competition states of local polar rather than relaxor.

Understanding the intricate correlation between structure and functional properties lies at the core of materials science. Crystal structure based on long-range periodicity with randomly distributed species provides excellent convenience in describing crystalline solids' properties. Yet, emerging functional crystalline materials usually harbor locally correlated disorder that is deviated from the global long-range average structure, such as that with respect to the composition or atomic displacements[1–4]. Critically, such structural disorder has a profound impact on their functional properties. For instance, the local Jahn-Teller distortion leads to a colossal magnetoresistive effect during the metal-insulator transition[5], and short-range chemical order alters the ionic transport properties and increases catalytic activity[6,7].

[1]Beijing Advanced Innovation Center for Materials Genome Engineering, University of Science and Technology Beijing, 100083 Beijing, China. [2]Department of Physical Chemistry, University of Science and Technology Beijing, 100083 Beijing, China. [3]School of Materials Science and Engineering, Beijing Institute of Technology, 100081 Beijing, China. [4]Chemical and Engineering Materials Division, Oak Ridge National Laboratory, Oak Ridge, TN 37831, USA. [5]Centre for Neutron Scattering, City University of Hong Kong, Kowloon, Hong Kong, China. [6]X-ray Science Division, Advanced Photon Source, Argonne National Laboratory, Argonne, IL 60439, USA. [7]Institute of Physics, University of Silesia, Katowice 40007, Poland. [8]Department of Materials Science and Engineering, Pennsylvania State University, University Park, PA 16802, USA. [9]These authors contributed equally: Hui Liu, and Xiaoming Shi. ✉e-mail: huiliu@ustb.edu.cn; hbhuang@bit.edu.cn; junchen@ustb.edu.cn

However, experimentally characterizing this local structural disorder, and correlating it with the advanced functions remain a big challenge.

Lead-based relaxor ferroelectric perovskite is a typical example of such material systems, which exhibit exceptional high electromechanical and dielectric properties, finding wide usage from sensing to actuators[8–10]. However, deciphering the origins of the enhanced properties has long been complicated by the complex local structural heterogeneity. In these materials, such as $(100-x)Pb(Mg_{1/3}Nb_{2/3})O_3–xPbTiO_3$ (PMN–$x$PT) solid solution, maximal piezoelectric response typically occurs at morphotropic phase boundary (MPB), where ferroelectric phases with different long-range crystal symmetries possess comparable energy. Common concepts based on long-range cooperative features, such as ease of polarization rotation via monoclinic phase[11,12], field-induced structural transition[13,14] etc., cannot completely rationalize the scenario that they present several times higher piezoelectric coefficients compared to classical ferroelectric counterparts.

The prime structural feature of lead-based relaxor ferroelectrics is the complex polar states arising from the local chemical composition inhomogeneity, which is generally argued to account for the high dielectric permittivity and characteristic dielectric relaxation[2,3,10,15–20]. Roughly, their superior piezoelectricity is ascribed to the much higher dielectric permittivity in contrast to classical ferroelectrics[21,22]. Different models[23–26] have been proposed to describe the structure and explain the role of the nanoscale polar entities in the relaxor dielectric signatures, such as the frequently-mentioned model of polar nanoregions (PNRs) or polar nanodomains (PNDs) embedded in a disordered matrix[27,28], the slush-like polar structure suggested by molecular dynamics simulations[2]. In these models, facilitated polarization rotation was identified as the predominant mechanism for piezoelectric enhancement. For instance, phase-field modeling shows that field-induced rotation of the PNRs inside the ferroelectric matrix dominates 50%-80% contribution to the piezoelectricity in single-domain lead-based relaxor crystals[20]. The strong coupling between lattice softening and PNRs has been demonstrated by X-ray/neutron scattering[29–31]. However, numerous relaxor ferroelectric systems with high dielectric permittivity are found to not process remarkable piezoelectricity. Meanwhile, the maximal piezoelectric response does not occur in composition with strong dielectric dispersion. It seems that the relaxor is not a generic feature for determining the piezoelectric performance, and what are the critical structural features for high piezoelectric performances still controversial.

Herein, we present how the hidden local atomic polar displacements evolve and assemble into global long-range structures across over realxor-ferroelectric states in the prototypical relaxor ferroelectric system of PMN-PT. 3D polar configuration spanning the phase diagram of PMN-$x$PT ($20 \le x \le 45$) were constructed by employing the newly advanced Reverse Monte Carlo (RMC) method to fit neutron-, X-ray-based total scattering, and extended X-ray absorption fine structure (EXAFS) data simultaneously. It demonstrates that the macroscopic relaxor behavior emerges from the local polar disordering in all directions, and the dielectric relaxation correlates strongly with the volume fraction of disorder polar-regions. The competing local polar order-disorder between different states with balanced local polar length and disorder leads to a flattening free-energy profile over a wide polar length confirmed by phase-field simulations. This critical feature is represented by the overlapping of long-range MPB and relaxor-ferroelectric boundary macroscopically, and thus gives rise to high piezoelectricity. Our work clarifies the generic feature required for high piezoelectric performances in relaxor ferroelectrics.

## Results

### Long-range average symmetry evolution
The useful functional properties of ferroelectric material are governed by their intrinsic polarization (**P**). In $ABO_3$ perovskite-type ferroelectrics, the **P** is comprised of the offset of $A$-site and $B$-site atoms from the center of their surrounding oxygen cage, termed as $A$-site and $B$-site polar displacement, respectively (Fig. 1a, b). It is well known that the **P** is strongly tied to crystal symmetries. The complicated average symmetry evolution of PMN-$x$PT ($10 \le x \le 50$) solid solution is demonstrated in Fig. 1c, as elucidated from refs. [32,33]. Notably, the average symmetry description of PMN-PT is subtle and still controversial, due to the existence of local chemical ordering, strong local static structural disorder[34–36]. The average symmetry of representative compositions ($x = 20$, 25, 30, 35, and 45) was identified by the high-resolution synchrotron X-ray powder diffraction, which offers super-high resolution ($\Delta Q/Q < 1.4 \times 10^{-4}$) and enables to detect the subtle peak splitting in MPB compositions (Supplementary Figs. 1–3). The structure refinement results demonstrate that the average symmetry evolves from rhombohedral ($R$, $R3m$) with polarization along $[111]_p$, to monoclinic ($M$), and then to tetragonal ($T$, $P4mm$) with polarization along $[001]_p$ with increasing Ti content (Fig. 1c, d). Monoclinic $M_C$ ($Pm$) with polarization lying on the $\{100\}_p$ mirror planes coexisted with comparable $T$ phase fraction are observed at 35PT. Monoclinic $M_B$ and $M_A$ is detected at 25PT and 30PT, respectively. These two types of $M$ have the same space group of $Cm$ with the polarization lying on the $\{110\}_p$ mirror plane, while $M_A$ acts as a structural bridge between $R$ and $T$, and the polarization in $M_B$ locates between $R$ and $O$ (Fig. 1c)[37]. The compositions near PMN end-number present interesting relaxor behavior, while the compositions near PT end-number possess conventional ferroelectric state. The intermediate compositions ($30 \le x \le 35$) located at the relaxor-ferroelectric boundary, concurrently with the so-called MPB, exhibits extraordinary electromechanical properties[38]. From this phenomenon, the enhanced piezoelectricity cannot be attributed to relaxor with prevailing disordered polar alone, the MPB with multiple long-range ferroelectric order states should be also considered.

### Composition-driven order-disorder feature along with continuous local atomic polar displacement rotation
Atomic pair distribution function (PDF) is a tool to go beyond the conventional diffraction analysis[39–42]. It provides quantitative information about local and sub-nanoscale structure. In this technique, PDF is Fourier transformed from the total scattering data, expressing the weighted distribution of atom pairs as function of the distance in real space. For the X-ray PDF, the O-related atomic pairs are relatively weak, while the distance between cations can be remarkably presented (Supplementary Fig. 4). Neutrons, however, enable capturing the $B$ site element-specific coordination and oxygen-related feature. Therefore, complementary neutron and X-ray total scattering measurements were performed to decode local structure and to interpret the compositional dependent electrical properties. The selected compositional dependence of neutron-PDFs (in the range of 1.5–4.3 Å) are shown in Fig. 2a, which carry the information of the nearest neighbor coordination in $ABO_3$ perovskite. Increasing the Ti content gives rise to a continuous change in PDF $G(r)$, but there is no sign of any abrupt structure change. The peaks located at 1.6–2.2 Å belong to the $B$-O pairs, which directly correlates to the $B$-site polar displacement. At least three signals at different distances corresponding to the overlap of three peaks can be observed in this region. The negative one at 1.75 Å mainly arises from Ti-O pair due to the negative neutron scattering length of Ti. The middle one at 1.95 Å and another one at 2.15 Å is mainly from the Nb-O and Mg-O pair, respectively, due to the larger ionic radius of Mg compared with Nb. Notably, this differentiation between different $B$-site cations cannot be refined in conventional diffraction studies using Bragg alone. Interestingly, the positions of the split $A$-O pairs appear to have changed slightly with composition, while the intensity varies significantly. This suggests the characteristic of

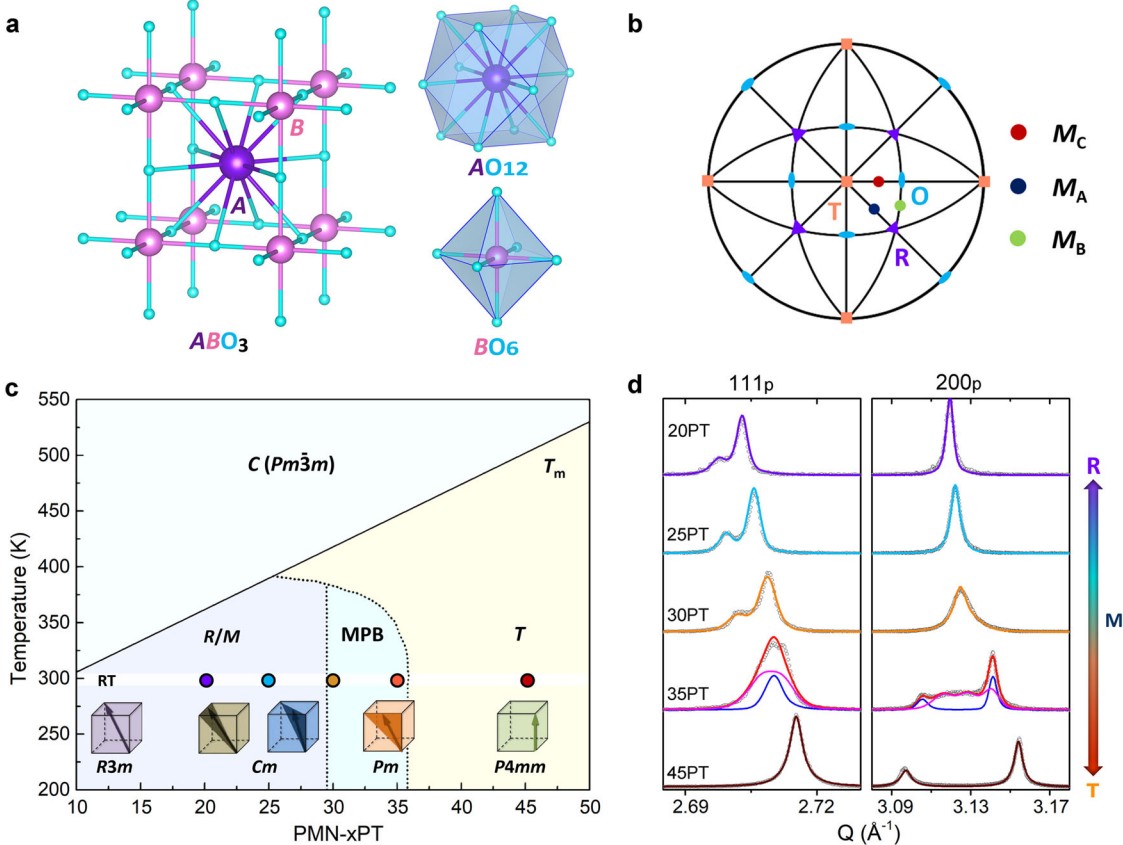

**Fig. 1 | Long-range average symmetry evolution. a** $ABO_3$ perovskite structure showing the surrounding oxygen cage of $A$-site and $B$-site cations. **b** Stereographic projection of a pseudocubic perovskite viewed along the $[001]_p$ direction based on the direction of polar vector. $M_A$, $M_B$ and $M_C$ denote different monoclinic structures. $T$, $R$, and $O$ specify tetragonal, rhombohedral, and orthorhombic structures, respectively. **c** Structural phase diagram for PMN–$x$PT with defined symmetries and morphotropic phase boundary. The dots mark the analyzed compositions at room temperature. **d** The results of Rietveld-refinement based on high-resolution synchrotron X-ray diffraction patterns. The circle point and solid line indicate the observed and fitted data, respectively. For 35PT, the blue and pink line show the individual peaks from $T$ and $M_C$, respectively.

order-disorder transition. Three main signals near 2.5 Å, 2.8 Å, and 3.2 Å are observed for Pb-O pairs. Consistent evolution is observed in X-ray PDF, while displaying more pronounced feature in $A$-$A$/$B$-$B$ and $A$-$B$ pairs (Supplementary Fig. 4).

In order to elucidate the complex nanoscale atomic correlations. Big-box modeling using an RMC algorithm by simultaneously fitting the neutron and X-ray total-scattering functions ($S(Q)$) and real space PDF, as well as Pb $L_{III}$-edge EXAFS data were performed (Supplementary Figs. 5–10). 3D atomic configurations with the approximately size of $64 \times 64 \times 64$ Å³ ($16 \times 16 \times 16$ perovskite unit cells with 20480 atoms) was built and both the atomic coordinates and swapped $B$-site atomic occupation was refined. The atomic polar displacement vector (**D**) was first extracted from the refined 3D atomic configuration. The mean atomic polar displacement length is plotted in Fig. 2b. It is evident that the Pb cation has relatively larger polar displacements of 0.34 Å compared with the $B$-site cations. This phenomenon is common for Pb-based perovskites due to the strong hybridization of Pb $6s$ with the O $2p$ states, and is often assigned as one of the main reasons for higher piezoelectricity. The atomic polar displacement length varies slightly with composition except for the Ti cation. Notably, $D_{Ti}$ presents more apparent change with composition, which increases with increasing PT content. These features suggest that a dominating order-disorder transition with respect to chemical composition occurs in Pb, Mg and Nb cations, while a mixture of order-disorder and displacive feature shows in Ti cation. This scenario is similar to the temperature-dependent behavior in PMN-25PT[43].

The probability density distributions of Pb atomic polar displacement are plotted (Fig. 2c). There is a strong effect of localized Pb polar displacement deviated from their average one. At 45PT, Pb polar displacement vectors are densely distributed along the $[001]_p$ direction, which agrees well with the average $T$ symmetry. However, evident disorder presents: isotropic disorder distribution occurs at $(110)_p$ plane, but preferential disorder distribution along $[100]_p$ direction is observed at $(101)_p$ plane, foreshowing the occurrence of monoclinic $M_C$ polarization in slightly lower PT content composition. With decreasing Ti content, the center of Pb polar displacement gradually deviates from the origin and toward to $[110]_p$ direction viewed from $[001]_p$ direction. Simultaneously, it is away from $[001]_p$ direction in $(101)_p$ plane. Such evolution will lead to a continuous polarization rotation. The most remarkable change occurs between 30-35PT, where show peak piezoelectric response, suggesting a strong correlation between local polar sate and piezoelectric properties. Starting from 30PT, the distributions of the local Pb polar displacement become broader with flat density maximums with further decreasing $x$. It suggests weaken long-range ferroelectric ordering and correlates with the emergence of relaxor behavior.

### Local polar order-disorder and polar rotation

In order to consider the local polar evolution comprehensively, the polar vector in each perovskite unit cell was calculated. The stereographic projections of local polarization viewed along $[001]_p$ and the corresponding average polar vector are shown in Fig. 3a. Evidently, a

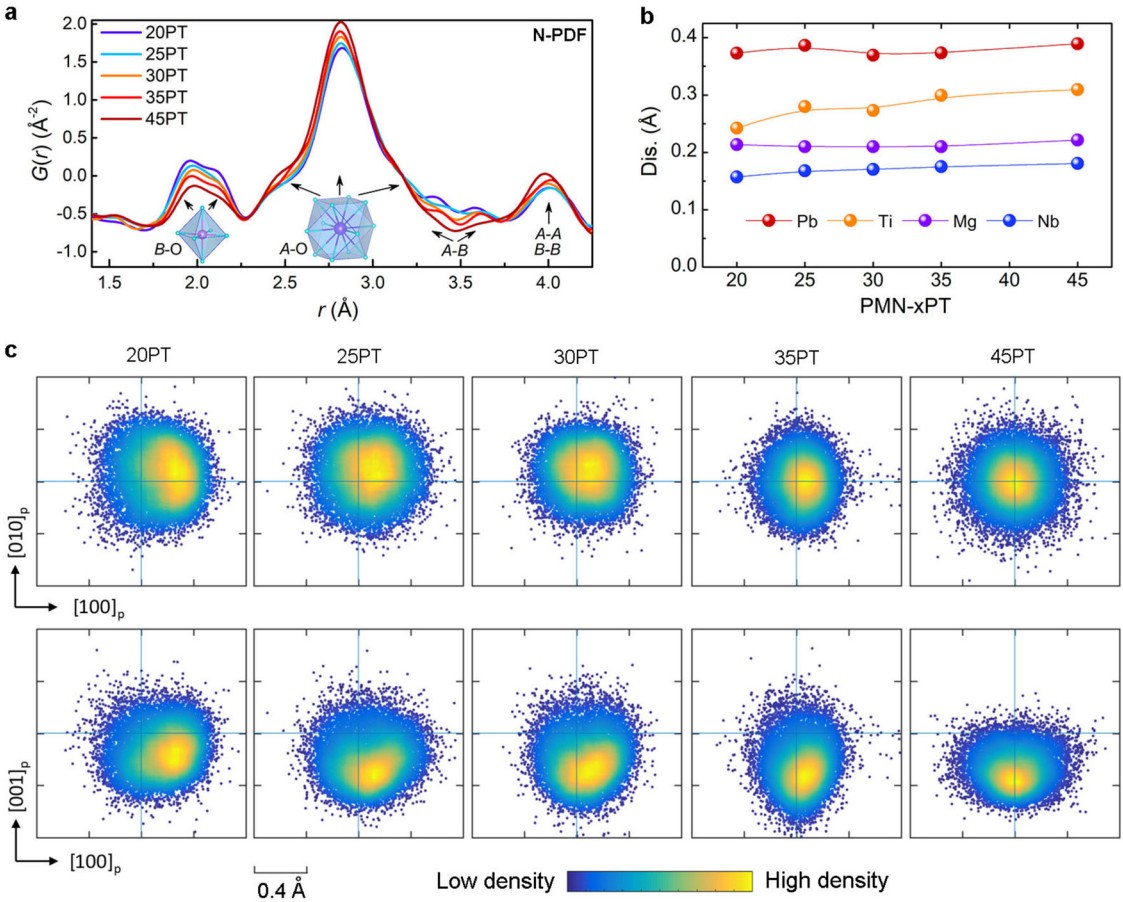

**Fig. 2 | Evolution of local polar displacements. a** Selected neutron PDF $G(r)$, showing the evolution of nearest atom-atom pairs. **b** Mean atomic polar displacement length determined from the center of respective oxygen polyhedron, extracted from the refined 3D atomic configurations. **c** Projections of the probability density distributions of Pb polar displacement. The guide line represents the origin without displacement.

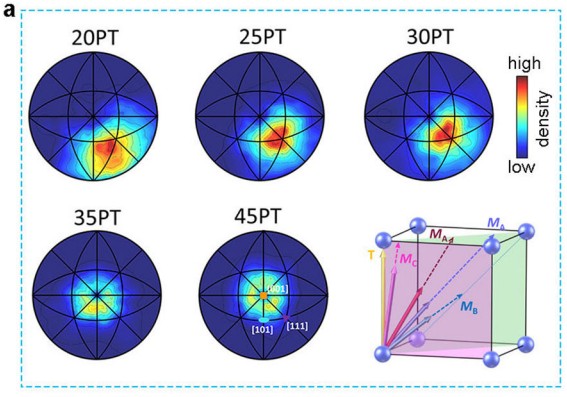

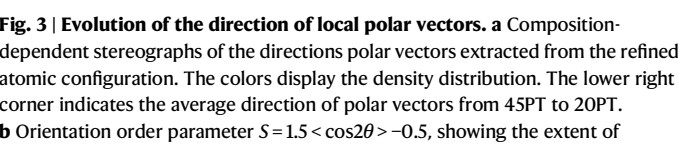

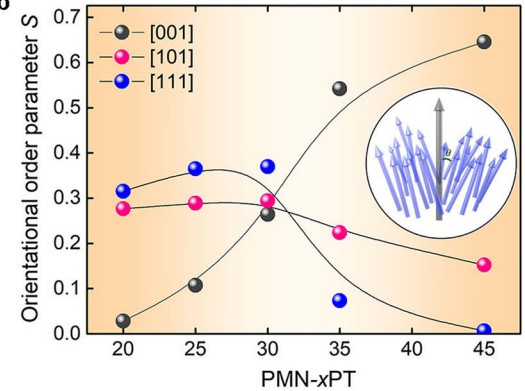

**Fig. 3 | Evolution of the direction of local polar vectors. a** Composition-dependent stereographs of the directions polar vectors extracted from the refined atomic configuration. The colors display the density distribution. The lower right corner indicates the average direction of polar vectors from 45PT to 20PT. **b** Orientation order parameter $S = 1.5 < \cos 2\theta > -0.5$, showing the extent of direction deviation of the polar vectors from the three representative directions, as indicates by the inset. Specifically, the value of 1 indicates perfect polar ordering, and the value of 0 denotes a complete polar disordering. It provides a quantitative description of the extent of randomness along the corresponding direction.

composition-driven continuous local polar rotation can be observed. A conical configuration with an angle deviating about 8° away from $[001]_p$ occurs in 45PT, and leads to an average polar vector along $[001]_p$. For 35PT, more polar vectors point between $O$ and $R$ with weak circular distribution around $[001]_p$ direction. It indicates that the multi-polar state emerges. The most significant changes are in 30-35PT. The local polar vector changes from the relatively concentrated states in 35PT to the multi-polar states with much disordered local $M_A$, $M_B$, and $M_C$ in 30PT. It suggests that high piezoelectricity emerges from the multi polar states with competitive order-disorder. Such polar state is represented by the overlapping of long-range MPB and relaxor-ferroelectric boundary macroscopically. With further decreasing the Ti content, the local polar vector evolves from average $M_A$ to $M_B$ with broadened distribution. It is consistent with the

macroscopically observed enhanced dielectric relaxation. Such a phenomenal link revealed here infers that the local polar order-disorder has a significant influence in the properties of PMN-PT.

Here the orientation order parameter ($S$) is defined as $S = 1.5 < \cos2\theta > -0.5$, to describe the alignment of the polarization direction[44]. $\theta$ is the angle between local polar vector and a given direction (as shown in the inset of Fig. 3b). Specifically, $S = 1$ indicates perfect order alignment (along the direction), $S = 0$ denotes a complete disorder alignment. $S$ can provide a quantitative description of the extent of randomness along the corresponding direction. Here, the order parameters $S$ along three main directions are calculated and plotted in Fig. 3b. Decreasing Ti content leads to the reduction of $S$ value along the $[001]_p$ direction, accompanied by firstly increasing and then decreasing value both along the $[111]_p$ and $[101]_p$ directions. Peak values of $S$ both along $[111]_p$ and $[101]_p$ directions are reached at around 30PT, where a critical composition region with relaxor-ferroelectric boundary. Interestingly, the comparable order parameters $S$ along different directions indicates a competing multi-polar state occurs between 30PT and 35PT, showing high piezoelectricity. This should be the critical feature responsible for the high piezoelectric properties. Notably, no evidence show the existence of long-range $O$ symmetry in the present studied compositions, but the relatively high local polar order occurs along $[101]_p$ direction, which is important to from the competing multi-polar state between $[001]_p$, $[101]_p$ and $[111]_p$. Such local competing polar states would be the reason why different long-range structure models were identified near MPBs of PMN-PT, such as the claimed $O$ symmetry[20,22]. Therefore, the local polar phenomenon commands the symmetry evolution in long-range scale. This scenario is similar to the recently demonstrated transformation between ordered and disordered phases in isometric pyrochlores[45,46] A high $S$ value along one direction signifies long-range ferroelectric ordering, as in 45 T. But the continuous reducing $S$ along all directions corresponds to an increasingly prominent relaxor feature, as changing from 30PT to 20PT. It needs to be mentioned that although lower $S$ values in all directions are presented in relaxor states, their piezoelectric response are not high. It is due to the much increased polar disorder and the loss of competitive behavior. The appearance of polar disorder solely cannot account for excellent electromechanical properties. This is why numerous relaxor ferroelectric systems do not have remarkable piezoelectricity. Instead, the critical structural feature for ultrahigh piezoelectricity in relaxors is the competing local multi-polar states. It needs to be emphasized that the competing local multi-polar states in PMN-PT not only arises from the the long-range multiphase coexistence, such as observed in conventional PZT ferroelectric system, but also comes from the local structural heterogeneity (Supplementary Fig. 11).

### Linking local polar disorder and length with high piezoelectricity

The characteristic of local polar vectors including the magnitude and direction disorder, and their development were further investigated. Generally, the magnitude of local polar vector is linked to the macroscopic polarization Fig. 4. To evaluate the direction disorder, the disorder parameter ($\xi$) is defined as $\xi = 1 - S$, which means the disorder with respect to the direction of the average polar vector were calculated. $\xi = 0$, means the polar vectors are completely disordered, suggesting an isotropic-polar configuration, $\xi = 1$ indicates a highly order state and corresponds to a strong anisotropic polar configuration. Increasing Ti content, the magnitude of local polar vector increases, while the disorder decreases. By comparing with the composition-dependent piezoelectric coefficients $d_{33}$, it can be concluded that high piezoelectricity emerges from the balanced local polar disorder and polar length. Polar configuration with large length but low disorder (as in 45PT), or small length but high disorder (as in 20PT) cannot give rise to high piezoelectricity. For example, the highest piezoelectric

response found at between 30PT with $P = 28\,\mu C/cm^2$ and $\xi = 0.5$ and 35PT with $P = 32\,\mu C/cm^2$, $\xi = 0.42$, not at the end points of 20PT where $\xi$ is large or at 45PT where $P$ is large. Interestingly, this finding can well reconcile the scenario that relaxor systems show much higher piezo-electric response compared to classical ferroelectrics with multiphase coexistence. As shown in Supplementary Fig. 11, for classical ferro-electric MPB composition of PZT53, the polar disorder is mainly from the long-range multiphase coexistence, and leads to a relatively low degree polar disorder ($\xi = 0.25$). While for PMN-PT system, the polar disorder is not only from the long-range multiphase coexistence, but also from the existence of relaxor feature, results in a higher degree polar disorder ($\xi = 0.5$ in 30PT). Therefore, the PMN-PT system displays much higher piezoelectric response ($d_{33} \approx 700$ pC/N) compared with PZT system ($d_{33} \approx 220$ pC/N), macroscopically. The competing of local polar order-disorder mechanism for piezoelectric enhancement is beyond the prevailing multiphase coexistence mechanism.

### Connection local polar inhomogeneity and properties

The 3D atomic polar configurations across over the phase boundary are displayed in the Fig. 5a. Local polar inhomogeneity exists in all compositions. Overall, the magnitude of polarization gradually increases, cooperating with an increasing coherence length from relaxor to ferroelectric state. Long-range ferroelectric ordering matrix coexisted with a small volume fraction of local disorder regions is observed in Ti-rich compositions, as in 45PT and 35PT. These local disorder regions are of a few unit-cell scales. While for $x \leq 30$, comparable volume fraction of different local polar entire-ties exists. Increasing volume fraction of local polar entireties with larger spatial size embedded in the log-range ferroelectric matrix, and disrupted the long-range polar ordering, results in increasingly apparent relaxor behavior. This scenario is consistent with the molecular dynamics simulation and TEM results[2,16], both of which show the existence of high-density, low-angle domain walls. Notably, the local polar-regions typically correspond to a small polarization, which could be arising from the differentiated polar feature in $B$ site atoms. The volume fraction of local disorder is highly correlated with the dielectric relaxation ($\Delta T_{max} = T_{max}@10^6$ Hz-$T_{max}@10^3$Hz). The volume fraction of local polar-regions increases rapidly between 30PT and 35PT, and this is just where the relaxor-ferroelectric boundary and conventional MPB are located. The favorable local polar clusters will smoothly connect different long-range polar states, and it's the reason for the piezoelectric enhancement. Previous studies indicated that the chemical short-range order of $Mg^{2+}/Nb^{5+}$ plays an important role in forming PNRs. The Ti, Mg, and Nb clustering was observed from the refined configuration in the low-Ti content, and the propensity of such a feature gradually diminishes with increasing Ti content (Supplementary Figs. 12 and 13). It would be the introduction of $Ti^{4+}$ that weakens the random electric fields, and balances the local Coulomb energy between $Mg^{2+}$ and $Nb^{5+}$. Comparing with Fig. 5a, it can be observed that a certain correlation exists between the chemical ordering and relaxor behavior, but no obvious spatial correlation to the PNRs can be observed. This scenario agrees well with the recent diffuse scattering results[3,25]. Note that for the composition with the maximal piezoelectric response, the $B$-site cations are randomly distributed. It seems that chemical order would be not required for high piezoelectricity.

### Discussion

The phase-field modeling was employed to elucidate the role of competing for polar order-disorder in the high piezoelectricity. In the modeling, local random fields are introduced to describe the relaxor feature[47]. Therefore, the randomness of polar vectors is characterized by a local random electric field variance. The composition-dependent local electric field variance with amplitude

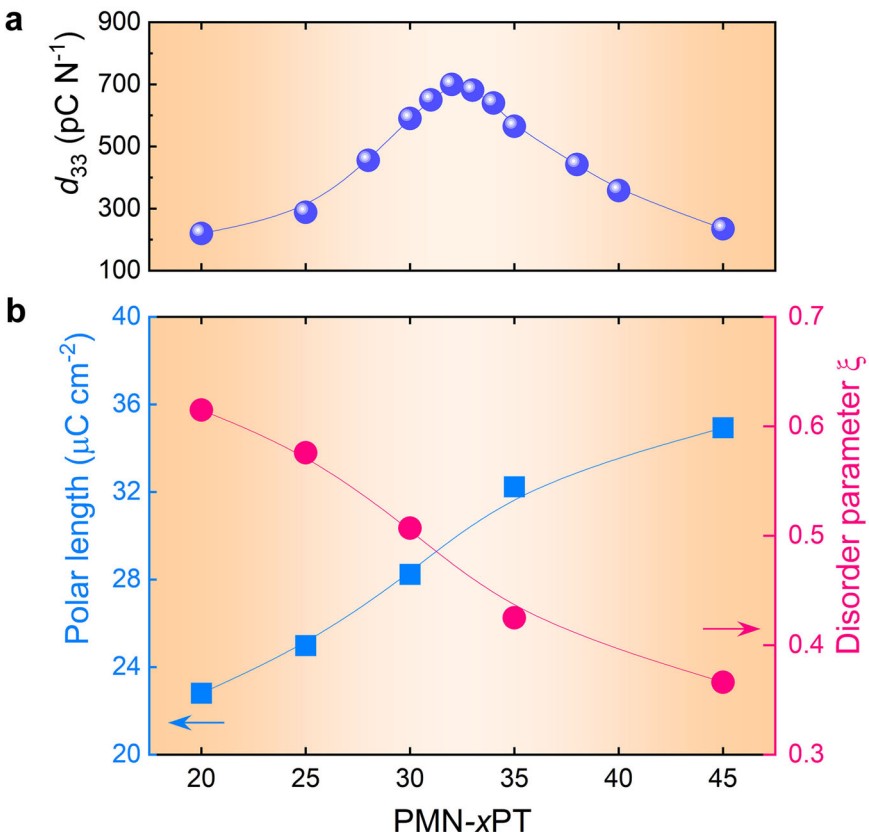

**Fig. 4 | Linking local polar disorder and length with piezoelectricity. a** The measured piezoelectric coefficient $d_{33}$, and **b** The calculated polar length and polar disorder parameter $\xi$ as function of compositions. The disorder parameter $\xi$ is defined as $1-S$. A higher value indicates higher degree of polar direction randomness.

positively related to the volume fraction of polar-disorder regions was introduced. One can see that the local random electric field deceases from 20PT to 40PT, which corresponds to the reduced polar disorder (Fig. 5b and Supplementary Figs. 14 and 15). As shown in Fig. 5c, the composition-dependent piezoelectric $d_{33}$ can be well reproduced from phase-field modeling. The corresponding schematic of Landau free-energy landscape is presented in Fig. 5d. The atomic polar configuration with a large polar length but weak disorder, such as, that presents in off-MPB Ti-rich compositions (40PT), possesses a deep energy barrier. On the other hand, the atomic polar configuration with a small polar length and high disorder occurs in near PMN compositions (20PT). In this state, although the local energy barrier is small, such a fatten free-energy profile appears in a narrow polar length range. Both two states command relatively low piezoelectricity. For the MPB composition (33PT), the competition between polar length and polar disorder occurs, which gives rise to a flat free-energy profile at a wide polar length range. The reduced polarization anisotropy enables favorable polarization rotation. Combined with large polar length and favorable polarization rotation, a significant field-induced polarization variation can be generated, which then ultimately results in high piezoelectricity (Supplementary Fig. 16). The polar domain structures are simulated (Fig. 5f and Supplementary Fig. 14), which are agree with the experimentally observations. Notably, such satisfactory evolution in domain structures and piezoelectric response cannot be obtained without introducing local random fields (Supplementary Fig. 15), suggesting the important role of local polar disorder. Notably, from the constructed 3D atomic configuration, the existing local disorder that highly correlated with relaxor feature, further reducing the polarization anisotropy as compared with the conventional piezoelectrics with long-range

multiphase coexistence feature, such as PZT. This is manifested as competing local polar vectors along three main directions with higher degree disorder, and can account for their exceptionally high piezoelectric response compared with conventional counterparts (Supplementary Fig. 11). In addition, prominent enhancement of piezoelectricity realized in lead-free based systems through rational chemical design, such as KNN and BT-based solid solutions, would mainly flat the free-energy profile through increasing the polar disorder[48,49]. For example, the slush polar state with multiphase coexistence boosted high piezoelectricity in KNN ceramic[50]. However, the fatten free-energy profile typically limited in narrow polar length range. Since the absence of hybridization between Pb $6s$ and O $2p$ in lead-free one leads to a relatively small polarization. This is the atomic-level reason that gaps still exists in piezoelectricity between lead-free and lead-based one. The competing local polar order-disorder mechanism should be applicable to other lead-based and lead-free relaxors, by considering their same physical principles, such as local chemical composition-driven local polar heterogeneity. Further, it is found that the hybridization of Pb $6s$ with O $2p$ states is important to stabilize the large local polar length under a high degree of direction disorder (Supplementary Fig. 17). Therefore, from the perspective of lead-free piezoelectric material design, introducing the strongly ferroelectric active ions, such as $Bi^{3+}$, $Sb^{5+}$, and $Sn^{4+}$ etc., would be beneficial to extend the local polar length. On the other hand, the long-range unit cell volume expansion would be necessary to introduce disorder into a solid solutions consisting of different ion sizes (Supplementary Fig. 3). Thus, it should be considered to increase the local polar disorder.

In summary, 3D atomic-level polar configurations were constructed based on the RMC refinement against the experimental

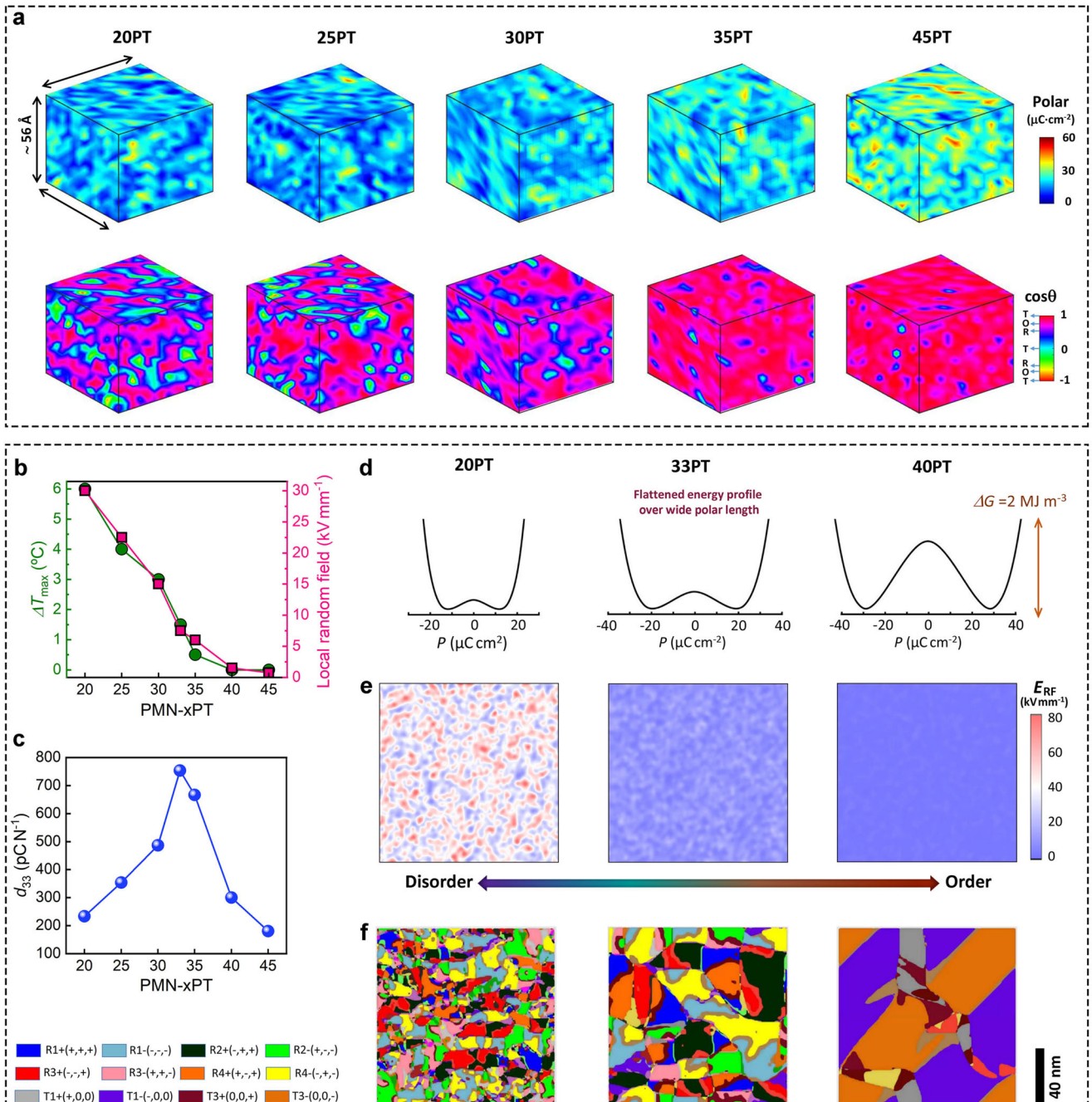

**Fig. 5 | Evolution of polar structures and connection with properties.**
**a** Evolution of polar structures obtained from refined 3D atomic configurations, showing the 3D distribution of the length and direction ($\cos\theta$) of unit-cell polar vectors. $\theta$ refers to the angle between the unit-cell polar vector and the $[001]_p$. **b** Composition-dependent dielectric relaxation $\Delta T_{max}$ and introduced local random field variance in phase field simulations. The $\Delta T_{max}$ is the peak shift of dielectric maximum from 1 kHz to 1000 kHz in temperature dependent-dielectric spectrum. **c** Estimated piezoelectric coefficient $d_{33}$ from phase field simulations. **d** The Landau energy profiles, **e** random field distribution ($E_{RF}$), and **f** simulated domain structures of representative compositions of 20PT, 33PT and 40PT. The effect of polar disorder is characterized by the local random field variance. A larger random field corresponds to higher degree of polar disorder.

neutron, X-ray total scattering, and EXAFS data, to extract the local structural inhomogeneity, and correlate the dielectric and piezoelectric properties of PMN-PT relaxor system. The local polar order-disorder feature along with continuous rotation commands the composition-driven global structure evolution. The relaxor behavior appears from the polar disordering in all directions, and the dielectric relaxation correlates strongly with the volume fraction of polar disordered regions. The competing local polar order-disorder between different states with balanced local polar length and direction randomness is the critical structural feature for the high piezoelectricity.

Such polar configuration leads to a fatten free-energy profile at a wide polar length range, which is also confirmed by phase-field simulations. The mechanism proposed in the present work is beyond the prevalent multiphase coexistence mechanism. Further, it is not limited to lead-based relaxor systems, but also can rationalize the current status in lead-free piezoelectric systems, such as KNN-based, BT-based solid solutions. Therefore, controlling and exploiting correlated local disordered states beyond the conventional "unit cell" concepts would be important to access functional responses inaccessible to conventional crystalline materials.

## Methods

See details of the methods part in the Supplementary Information files.

## Data availability

Relevant data supporting the key findings of this study are available within the paper and the supplementary information file. Source data are provided as a Source Data file. Source data are provided with this paper.

## Code availability

The RMCProfile software (executables) can be downloaded from www. rmcprofile.org.

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

## Acknowledgements

This work was supported by the National Natural Science Foundation of China (Grant Nos. 22235002, and 22075014), the Fundamental Research Funds for the Central Universities, China (Grant No. 06500162), the China Postdoctoral Science Foundation (BX20200044, 2020M680344). A portion of this research used resources at the Spallation Neutron Source, a DOE Office of Science User Facility operated by the Oak Ridge National Laboratory. The use of the Advanced Photon Source at Argonne National Laboratory was supported by the U.S. Department of Energy, Office of Science, Office of Basic Energy Sciences, under Contract No. DE-AC02-06CH11357.

## Author contributions

H.L., J.C. conceived the idea of this work. H.L., Y.Y. and HuajieL. fabricated the solid solutions and performed the electrical properties measurements. H.L. performed structural refinements and data analysis with the assistance of Q.L. and Y.Z.. Y.Z. and J.N. collected neutron total scattering data, S.K. collected EXAFS data, H.L. and Y.R. collected X-ray total scattering data. X.S. and H.H. performed the phase-field simulations with the assistance of L.Q.C. H.Q. and K.R. participated the discussions of piezoelectric mechanism. All authors discussed the results and commented on the manuscript. H.L., X.R.X. and J.C. guided the projects.

## Competing interests

The authors declare no competing interests.
