## [Peer Review File · Nature Communications]

Emergence of high piezoelectricity from competing local polar order-disorder in relaxor ferroelectricsREVIEWER COMMENTS

Reviewer #1 (Remarks to the Author):

This work constructed the 3D atomic-level polar configurations for PMN-PT relaxor system based on the RMC refinement against the experimental neutron, X-ray total scattering, and EXAFS data. It illustrated that the relaxor behavior appears from the polar disordering in all directions, and the dielectric relaxation correlates strongly with the volume fraction of polar disordered regions. The present work will be helpful for understanding of the local polar and order-disorder characteristics in relaxor ferroelectrics. There are some comments or questions should be clarified.

- (1) In Fig.3(a), the directions of polar vectors for 20PT and 45PT are significantly different, but the piezoelectric coefficients d_{33} of these two compositions in Fig. 4(a) have no much difference, what is the reason?
- (2) According to the explanation in Fig. 3(b), the “order” refers to the principal polarization direction, the “disorder” refers to other polarization directions with random distributions, like domain variants of other phase structures. The high piezoelectric properties may come from the multiphase coexistence near the MPB, which has also been revealed by previous studies. So, what is the difference between the “order-disorder” competing and the multiphase coexistence mechanism for explanation the outstanding piezoelectric properties of relaxor ferroelectrics?
- (3) What are the specific components of the materials corresponding to each energy barrier in Figure 5c?
- (4) The description of Figure s3 needs to be more specific. What do the abscissa and ordinate represent?
- (5) From the phase field simulation results in Supplementary Material S10, the R domain structure are dominant at 20PT, while the T phase domain structure are dominant at 40PT. Near the MPB, the relaxor ferroelectric with high d_{33} is in a state of two-phase coexistence. What is the competition between order and disorder?
- (6) Can the phase field simulations give more results to support the author's point?
- (7) The author believes that the competing of the local polar order-disorder between different states and the balanced local polar length and direction randomness are the critical structural feature for the high piezoelectricity. Can these features be quantified? What will the combinations make the piezoelectricity increase?

Reviewer #2 (Remarks to the Author):

Referee report

Titled “Emergence of high piezoelectricity from competing local polar order-disorder in relaxor ferroelectrics

By Hui Liu, Xiaoming Shi, Yonghao Yao, Huajie Luo, Qiang Li, Houbing Huang, He Qi, Krystian Rolder, Yuanpeng Zhang, Yang Ren, Shelly D. Killy, Jeorg C. Neuefeind, Long-Qing Chen, Jun Chen

XAFS, Neutron PDF, and X-ray PDF are used to refine the nanoscale structure of PMN-PT solid solutions. In lead-based perovskite of PMN-PT, the PbO_{12} unit is mainly responsible for the polarization, and the BO_6 octahedral unit is responsible for the randomness of the polarization.

Regarding PDF data, it is not difficult as an experiment conducted on SNS and APS. Also, the conditions for data reduction are described in the supplemental part, so it is reliable data. However, it is not shown whether the $G(r)$ in Fig. S2 and S4 are resolution-corrected data corresponding to the model assumed in RMC.

Authors succeeded in quantifying the disorder, which is difficult to define as a structural parameter, and showing its relationship with piezoelectricity. Although MD calculation is usually used for verification of RMC modeling, phase-field simulations have also successfully verified the disorder parameter.

It is a great achievement that the authors were able to construct a structural model that can explain macropolarization in a mere sub-nanometer region, although composition dependence of the structure is hardly observed in the region below the unit cell.

Authors have also succeeded in extracting the disorder parameters precisely in RMC modeling, which tends to prefer dissipative structures.

Overall, the explanation of the figure caption is insufficient, and it is not clear what the insets, arrows, and labels in the figure are intended to indicate, so corrections are necessary. In addition, I have noted below the points that mislead the reader, so please take this into consideration when making revisions.

1. XAFS, Neutron PDF, and X-ray PDF are used as experimental data for constructing the local structure model. Of these, strictly speaking, XAFS and N-PDF cannot obtain static real space information. There is a dynamic structural contribution because the energy of the probing source is very close to the vibrational energy of the lattice. The contribution of the dynamic structure to the three kinds of real space information is different, but it is necessary to explain how they were converged to the static structure model.

2. Connecting lattices with different polarizations distorts the lattice and destabilizes the structure. What is the mechanism by which disordered polarization is stabilized? You mention the polarization due to the large Pb displacement due to the hybridization of Pb and oxygen, but can it be extracted from the model structure that Pb inhibits the polarization ordering?

Fig. 3

a

It can be clearly seen that the monoclinic structure appears in the disorder phase. The correspondence between 25PT M_{B} , 30PT M_{A} , and 35PT M_{C} is slightly inconsistent between Fig. 3a and Fig 1b.

b

In Fig. 1c, the monoclinic phase is shown to appear at the phase boundary between the rhombohedral and tetragonal phases, thus the importance of the orthorhombic polarization should be explained.

Fig. S2

Volume expansion is necessary to introduce randomness into a mixture of materials consisting of different ion sizes. Is there any indication of this in the average structure obtained by Rietveld analysis?

Fig. S7 Pb-L3 XAFS

EXAFS oscillations are difficult to observe in PMN-rich XAFS due to structural disorder and phase difference between Mg and Nb. Absorption spectra should be displayed to examine

data quality.

pS4L64

The minus of “-1/6” in Eq (3) is hard to see.

Reviewer #3 (Remarks to the Author):

This manuscript studies the origin of high piezoelectricity in the widely used PMN-PT relaxor ferroelectrics. The authors applied both experimental and simulation tools, including those X-ray scattering, neutron scattering, and phase-field simulations, and concluded that the origin of high piezoelectricity comes from the balanced local polar length and direction randomness. This is an exciting research topic. For a long time, there have been controversial arguments about the major contribution to piezoelectricity in relaxor ferroelectrics. In particular, many models to describe the highest piezoelectricity in MPB have been proposed, but they all stand at different angles with ambiguous conclusions. Electron microscopy has attracted a lot of attention recently to address this issue, but the nature of which focuses on local regions that may not represent the origin as a whole. The current study provides an excellent platform and a new attempt to settle the above-mentioned argument by using various advanced scattering techniques. The manuscript is well-organized and technically sound. I would be more than happy to recommend its publication in Nature Communications upon receiving clarifications from the authors on the following issues.

While both the 30PT and 35PT lie in MPB, their synchrotron X-ray diffraction patterns are completely different. In particular, three curves were demonstrated in Fig. 1c with different peak values and slight peak shifts. To what extent does this represent and what is the reason behind need further clarification.

The nearest neighbour coordination is changed upon changing the Ti content. However, the $G(r)$ of the Ti-O remain almost the same while the position and $G(r)$ of the Nb-O, Mg-O, and PbO pairs are significantly altered with the change of Ti content. The authors need to provide a detailed explanation of this and the related implication.

Could the conclusion be extended to other lead-based relaxor ferroelectric systems or even lead-free relaxor ferroelectric systems? I hope the authors can comment on this point.

Dear reviewers,

Thank you so much for spending your time reading our manuscript and providing insightful comments. In the following, we present our response to your comments, point by point. Based on your comments and suggestions, we have further improved the manuscript. We hope that these changes adequately address the concerns raised. We have highlighted all the changes within the manuscript in the red-colored text.

Reviewer #1 (Remarks to the Author):

This work constructed the 3D atomic-level polar configurations for PMN-PT relaxor system based on the RMC refinement against the experimental neutron, X-ray total scattering, and EXAFS data. It illustrated that the relaxor behavior appears from the polar disordering in all directions, and the dielectric relaxation correlates strongly with the volume fraction of polar disordered regions. The present work will be helpful for understanding of the local polar and order-disorder characteristics in relaxor ferroelectrics. There are some comments or questions should be clarified.

Reply: We thank reviewer for the positive opinion about our work.

Comment 1: In Fig. 3(a), the directions of polar vectors for 20PT and 45PT are significantly different, but the piezoelectric coefficients d_{33} of these two compositions in Fig. 4(a) have no much difference, what is the reason?

Reply 1: Thank for the good comment. As the reviewer pointed out, the directions of polar vectors in 20PT and 45PT are significantly different but their piezoelectric coefficients d_{33} are close (20PT: $d_{33} = 220$ pC/N; 45PT: $d_{33} = 235$ pC/N). As shown in Fig. 3, for 45PT, the polar vectors are mainly along $[001]_c$, with high order (disorder parameter $\xi = 0.38$) and relatively large polar length ($P = 35$ $\mu\text{C}/\text{cm}^2$), while for 20PT, the polar vectors shows much disorder (disorder parameter $\xi = 0.6$) and relatively small polar length ($P = 23$ $\mu\text{C}/\text{cm}^2$). Based on our results, the piezoelectricity is highly correlated with the local polar length and direction randomness. Local polar configuration with large polar length but low directions disorder (such as 45PT: $P = 35$ $\mu\text{C}/\text{cm}^2$, $\xi = 0.38$), or small length but high direction disorder in local polar vectors (such as 20PT: $P = 23$ $\mu\text{C}/\text{cm}^2$, $\xi = 0.6$) cannot give rise to high piezoelectricity. In other words, the relatively low piezoelectric coefficients d_{33} in both 20PT and 45PT is caused by two different scenarios, the former possesses high disorder but small length in local polar vectors; while the latter possesses less disorder but large length in local polar vectors (as shown in Fig.

4 and Supplementary Fig. 16). The competing local polar order-disorder between different states with balanced local polar length and direction randomness is critical for high piezoelectricity. Therefore, it is reasonable that both 20PT and 45PT present relative low piezoelectric coefficients d_{33} .

Revision: We have added some discussions in main text.

Page 19: “Polar configuration with large length but low disorder (as in 45PT), or small length but high disorder (as in 20PT) cannot give rise to high piezoelectricity.”

Comment 2: According to the explanation in Fig. 3(b), the “order” refers to the principal polarization direction, the “disorder” refers to other polarization directions with random distributions, like domain variants of other phase structures. The high piezoelectric properties may come from the multiphase coexistence near the MPB, which has also been revealed by previous studies. So, what is the difference between the “order-disorder” competing and the multiphase coexistence mechanism for explanation the outstanding piezoelectric properties of relaxor ferroelectrics?

Reply 2: Thank for the good comments. We totally agree that the multiphase coexistence can lead to the increasing of local polar direction randomness (disorder). On the other hand, we would like also to point that local polar direction randomness (disorder) can occur in single phase, as observed in 45PT, due to the existed chemical composition disorder. For relaxor ferroelectric systems, the polar disorder is not only from the multiphase coexistence (polar direction in average structures, such as R , T , O , M), but also comes from the relaxor state (local structure inhomogeneity: randomness from the polar direction deviated from average structures). This critical feature is represented by the overlapping of long-range MPB and relaxor-ferroelectric boundary macroscopically.

As a comparison, we have conducted neutron-PDF measurements of conventional $\text{Pb}(\text{Zr},\text{Ti})\text{O}_3$ (PZT) ferroelectrics including the T composition of PbTiO_3 (PT), R composition of $\text{PbZr}_{0.65}\text{Ti}_{0.35}\text{O}_3$ (PZT65) and MPB composition of $\text{PbZr}_{0.53}\text{Ti}_{0.47}\text{O}_3$ (PZT53). The MPB composition of PZT53 presents multiphase coexistence, but do not have relaxor feature. As shown in Fig. R1(a), for PZT system, the orientational order parameter S of local polar vectors along [001] direction decreases, accompanied by the increasing in S along [111] direction from T -to-MPB-to- R . While for the PMN-PT relaxor system, the behavior are completely different. With composition from T approaching MPB, the orientational order parameter S along [001] direction reduced, and increase along [111] direction. While the order parameter S along three directions ([001], [110], and [111]) decreases from MPB-to- R , which is completely different with the PZT system. This arises from the relaxor feature.

Fig. R1. Comparison of composition-dependent orientation order parameter S between the conventional ferroelectric system of PZT (a), and the relaxor ferroelectric system of PMN-PT (b).

Specifically, for MPB composition, in PZT53, the polar disorder is from the multiphase coexistence, and leads to a less polar disorder (disorder parameter $\xi = 0.25$). In PMN-PT system, the polar disorder is not only from the multiphase coexistence, but also from the existence of relaxor feature (disorder parameter $\xi = 0.5$ in 30PT). The existence of relaxor feature leads to higher degree disorder polar direction in PMN-30PT compared with conventional PZT53. Macroscopically, the PMN-PT system displays much higher piezoelectric response ($d_{33} \approx 700$ pC/N) compared with PZT ($d_{33} \approx 220$ pC/N). Therefore, the mechanism of competing of local polar order-disorder is beyond the multiphase coexistence. The multiphase coexistence cannot rationalize the scenario that relaxor systems present several times higher piezoelectric coefficients compared to classical ferroelectric counterparts.

Revision: We have added all these in the Supplementary Fig. 11, and some discussions in main text.

Page 9: “It needs to be emphasized that the competing local multi-polar states in PMN-PT not only arises from the the long-range multiphase coexistence, such as observed in conventional PZT ferroelectric system, but also comes from the local structural heterogeneity (Supplementary Fig. 11).”

Page 11: “Interestingly, this finding can well reconcile the scenario that relaxor systems show much higher piezoelectric response compared to classical ferroelectrics with multiphase coexistence. As shown in Supplementary Fig. 11, for classical ferroelectric MPB composition of PZT53, the polar disorder is mainly from the long-range multiphase coexistence, and leads to a relatively low degree polar disorder ($\xi = 0.25$). While for PMN-PT system, the polar disorder is not only from the long-range multiphase coexistence, but also from the existence of relaxor feature, results in a higher degree polar disorder ($\xi = 0.5$ in 30PT). Therefore, the PMN-

PT system displays much higher piezoelectric response ($d_{33} \approx 700$ pC/N) compared with PZT system ($d_{33} \approx 220$ pC/N), macroscopically. The competing of local polar order-disorder mechanism for piezoelectric enhancement is beyond the prevailing multiphase coexistence mechanism.”

Comment 3: What are the specific components of the materials corresponding to each energy barrier in Figure 5c?

Reply 3: Thank for the good comment. The energy barrier shown in Fig. 5c (now, Fig. 5d in revised manuscript) is corresponding to the PMN-20PT, PMN-33PT, and PMN-40PT. We have added the composition in the figure caption.

Comment 4: The description of Figure S3 needs to be more specific. What do the abscissa and ordinate represent?

Reply 4: Thank for the suggestion. We have replotted and revised the captions of this Figure. The updated Fig. S3 (now, Supplementary Fig. 4 in revised manuscript) is pasted here for reference.

Supplementary Fig. 4. X-ray atomic pair distribution functions (PDFs) $G(r)$, determined from X-ray total scattering in the interatomic distance r range of 1–10 Å, showing the evolution of short-range atom-atom correlations as a function of composition in PMN- x PT. The abscissa represents the interatomic distance, and ordinate represent the intensity of the reduced pair distribution function $G(r)$.

Comment 5: From the phase field simulation results in Supplementary Material S10, the R domain structure are dominant at 20PT, while the T phase domain structure are dominant at 40PT. Near the MPB, the relaxor ferroelectric with high d_{33} is in a state of two-phase coexistence. What is the competition between order and disorder?

Reply 5: Thank for good comments. We are sorry for the unclear description of the phase field simulation results. Indeed, the R domain structure are dominant at 20PT, while the T phase domain structure are dominant at 40PT, and near the MPB, the relaxor ferroelectric with high d_{33} is in a state of two-phase coexistence in phase-field simulations. We would like to point that,

in the phase-field simulations, a random field was introduced to account for the polar disorder according to the relaxor feature. Random electric field is commonly used to simulate the relaxor behavior in relaxors^{1,2}. During the phase-field simulations, the composition-dependent local electric field variance with amplitude positively related to the volume fraction of polar-disorder regions was introduced (Fig. 5b and Supplementary Fig. 14). Thus, the random field variance decreased from 30 kV/mm in 20PT to 0 kV/mm in 45PT. As shown in Supplementary Fig. 15, the domain structures with/without introducing random electric field (disorder) were simulated. One can clearly see that the domain structures are completely different between these with introduction of random electric field (consider as disorder) and these without. The local polar direction randomness (random electric field) can break the long-range ferroelectric order. Thus, the introduction of random electric field reduces the domain size and make the domain wall dispersion, all these are important polar characteristic of relaxors and are more consistent with what's actually observed. The competing local polar order-disorder between different states with balanced local polar length and direction randomness exists in the region of $x = 0.33$, and high piezoelectric d_{33} emerges.

Supplementary Fig. 14. (a) Domain structures calculated from phase-field simulation. The main domain structures are represented by different colors. (b) The local random field distribution. The effect of polar disorder is characterized by the local random field variance. A larger random field corresponds to higher degree of polar disorder. The composition-dependent local electric field variance with amplitude positively related to the volume fraction of polar-disorder regions was introduced. Correspondingly, the random field variance decreased from 30 kV/mm in 20PT to 0 kV/mm in 45PT. The domain size decreases, and the domain structures change obviously from 40PT to 20PT.

Supplementary Fig. 15. Comparison of the domain structures calculated from phase-field simulation between with (a) and without (b) introducing random electric fields. In the phase-field simulations, random electric field are commonly used to simulate the relaxor behavior in relaxors due to the chemical disorder. Without introducing random electric field, one can not get the correct Landau energy profiles, and thus unable to get the correct domain structures and piezoelectric response. One can clearly see that the domain structures are completely different between these with introduction of random electric field (consider as disorder) and these without. The local polar direction randomness (random electric field) can break the long-range ferroelectric order. Thus, the introduction of random electric field reduces the domain size and make the domain wall dispersion, these important polar characteristic are more consistent with what's actually observed.

Revision: We have added all these in the Supplementary Fig. 14 and Supplementary Fig. 15, in the Supplementary information and revised the phase-field phase field simulation part in main text (Page 13-14).

Comment 6: Can the phase field simulations give more results to support the author's point?

Reply 6: Thank for good comments. We would like to point that the local polar disorder-order effect was involved in calculating the piezoelectric coefficients during phase field simulations, and the calculated composition-dependent d_{33} is consistent well with the measured one. The energy barrier in Fig. 5d can also support our results. On the other hand, we add the domain structure and local random field distribution (disorder) evolution to support the completing local polar disorder-order mechanism. As shown in revised Fig. 5. The local polar disorder-order feature are shown more clearly.

Figure 5. (a) Evolution of polar structures obtained from refined 3D atomic configurations, showing the 3D distribution of the length and direction ($\cos\theta$) of unit-cell polar vectors. θ refers to the angle between the unit-cell polar vector and the $[001]_p$. (b) Composition-dependent dielectric relaxation ΔT_{max} and introduced local random field variance in phase field simulations. The ΔT_{max} is the peak shift of dielectric maximum from 1 kHz to 1000 kHz in temperature dependent-dielectric spectrum. (c) Estimated piezoelectric coefficient d_{33} from phase field simulations. (d) The Landau energy profiles, (e) random field distribution, and (f) simulated domain structures of representative compositions of 20PT, 33PT and 40PT. The effect of polar disorder is characterized by the local random field variance. A larger random field corresponds to higher degree of polar disorder.

Comment 7: The author believes that the competing of the local polar order-disorder between different states and the balanced local polar length and direction randomness are the critical structural feature for the high piezoelectricity. Can these features be quantified? What will the combinations make the piezoelectricity increase?

Reply 7: Thank for the good comment. Based on the recent advanced nanostructural

characterization method, the local polar length and direction randomness the can be quantitatively extracted from the reconstructed 3D atomic configurations (note that these critical local structural feature cannot be quantified by other experimental characterization techniques). For example, the highest piezoelectric response found at between 30PT with $P = 28 \mu\text{C}/\text{cm}^2$ and $\xi = 0.5$ and 35PT with $P = 32 \mu\text{C}/\text{cm}^2$, $\xi = 0.42$, not at the end points of 20PT where ξ is large or at 45PT where P is large. In order to understand better what we found, we draw a simplified schematic diagram (Fig. R2). The more disorder means a lower energy potential barrier between different polar states. Therefore, under applied electric field easier for polar rotation. On the other hand, if the local polar length is large, it means that the electric field causes larger polar variation, and thus gives rise to high piezoelectricity. We have added these in the main text.

Fig. R2. Schematic diagram of high piezoelectricity emerges from the competing of the local polar order-disorder between different states and the balanced local polar length and direction randomness.

On the other hand, inspiring by the reviewer, and considering the relationship $d = 2PQ\epsilon$, we try to put the polar length (P) and disorder parameter (ξ) together ($\xi \times P^n$). It seems the similar trend between the compositional dependent d_{33} and $\xi \times P^n$ occurs (Fig. R3). While the relationship need to be confirmed by more piezoelectric systems and the specific physical meaning of $\xi \times P^n$ will be considered in our further study.

Fig. R3. The possible relationship between the $\xi \times P^n$ and the piezoelectric d_{33} , which indicates the the competing of the local polar order-disorder between different states and the balanced local polar length and direction.

Revision: We have added all these in the Supplementary Fig. 16, in the Supplementary information and added corresponding discussion main text.

Reviewer #2 (Remarks to the Author):

XAFS, Neutron PDF, and X-ray PDF are used to refine the nanoscale structure of PMN-PT solid solutions. In lead-based perovskite of PMN-PT, the PbO_{12} unit is mainly responsible for the polarization, and the BO_6 octahedral unit is responsible for the randomness of the polarization. Regarding PDF data, it is not difficult as an experiment conducted on SNS and APS. Also, the conditions for data reduction are described in the supplemental part, so it is reliable data.

Reply: We greatly appreciate your positive overall comments.

However, it is not shown whether the $G(r)$ in Fig. S2 and S4 are resolution-corrected data corresponding to the model assumed in RMC.

Reply: Thank for the good comments. During RMC fitting, the $G(r)$ are considered the resolution effect. The NIST Si SRM powder total scattering data were collected to identify the instrument resolution in both X-ray and neutron total scattering measurements³. The calculated total-scattering data were corrected for the instrument resolution in both reciprocal and real spaces. We have added these missed information in the Supplementary Information, and the figure caption of Supplementary Figs. 6 and 7.

Authors succeeded in quantifying the disorder, which is difficult to define as a structural parameter, and showing its relationship with piezoelectricity. Although MD calculation is usually used for verification of RMC modeling, phase-field simulations have also successfully verified the disorder parameter. It is a great achievement that the authors were able to construct a structural model that can explain macropolarization in a mere sub-nanometer region, although composition dependence of the structure is hardly observed in the region below the unit cell. Authors have also succeeded in extracting the disorder parameters precisely in RMC modeling, which tends to prefer dissipative structures.

Reply: We sincerely thank the reviewer for the comments and recognition of our study.

Overall, the explanation of the figure caption is insufficient, and it is not clear what the insets, arrows, and labels in the figure are intended to indicate, so corrections are necessary. In addition, I have noted below the points that mislead the reader, so please take this into consideration when making revisions.

Reply: Thank for the good comments. We have revised the figure captions to show the figure clearer.

Comment 1: XAFS, Neutron PDF, and X-ray PDF are used as experimental data for constructing the local structure model. Of these, strictly speaking, XAFS and N-PDF cannot obtain static real space information. There is a dynamic structural contribution because the energy of the probing source is very close to the vibrational energy of the lattice. The contribution of the dynamic structure to the three kinds of real space information is different, but it is necessary to explain how they were converged to the static structure model.

Reply: Thank for the good comments. We totally agree with the review's point that the contribution of the dynamic structure to the three kinds of real space information is different. Here, we would like to explain the concern by the following:

(i) The reverse Monte Carlo (RMC) routine is a pure data-driven approach, meaning the final model obtained via the combined fitting is a compromise of structural aspects associated with different types of datasets. In another word, the convergence of the model is purely determined by the overall agreement between the calculated patterns and the experimental ones. Specifically, for those datasets containing more significant dynamic contribution (e.g., neutron total scattering, as pointed out by the referee, due to closer energy of thermal neutrons as compared to the low energy lattice vibrations), the corresponding structure solution would contain more dynamic characteristics. On the contrary, the high energy of X-ray would lead to relatively weaker dynamic contribution in the data set and therefore the correspondingly driven structure model would possess more static characteristics. In practical RMC modeling, the contribution from different datasets (thus the driving force towards different structure characteristics) is controlled by the weight factor and the golden rule, which is what we have been following for our RMC modeling, is that the weight for different datasets should be balanced to guarantee the end result not being biased.

(ii) In the RMC structure modeling, the dynamic and static aspects of the structure is not decoupled – decoupling. The static and dynamic structural aspects is also not the focus of current report and the discussion presented in current report is not based on the distinction between the static and dynamic characteristics of the structure. Furthermore, the discussion presented in current manuscript is not based on the distinction between the dynamic and static factors with respect to the local structural distortion (or, polarization, etc.).

Comment 2: Connecting lattices with different polarizations distorts the lattice and destabilizes the structure. What is the mechanism by which disordered polarization is stabilized? You mention the polarization due to the large Pb displacement due to the hybridization of Pb and oxygen, but can it be extracted from the model structure that Pb inhibits the polarization ordering?

Reply 2: Thank you for the good comments. The local polar heterogeneity is mainly from the local chemical composition fluctuations that different *B*-site cations (Ti^{4+} , Mg^{2+} , and Nb^{5+}). As widely known, the Pb^{2+} is the strongly ferroelectric active cation. The Ti^{4+} is also strongly ferroelectric active *B*-site cations, Nb^{5+} is moderately, while Mg^{2+} is weakly ferroelectric active cation. Due to the strong hybridization between Pb 6*s* with O 2*p* states, the Pb possesses large polar displacement. The PbTiO_3 (PT) has strong long-range ferroelectric polar ordering (as shown in Fig. R1(a), order parameter *S* near 1, which corresponds to completely polar order along [001]). While for PMN ($\text{Pb}(\text{Mg}_{1/3}\text{Nb}_{2/3})\text{O}_3$), the long-range polar order is broken, short-range polar order exists. We totally agree that the Pb is important to stabilize the disorder polar. Besides, the ferroelectric active *B*-site cations also play some role. Because the Pb displacement is affected by the local chemistry.

We try to extract the charge density distribution from the model structure, however the size of the model is too big to extract all charge density distribution in the model. We extract the charge density distribution in the part of the structural model. As shown in Supplementary Fig. 17, we can see the strong hybridization between Pb 6*s* with O 2*p* states, and inhibiting the polarization ordering. These are added in the Supplementary Fig. 17.

Supplementary Fig. 17. (a) One of the RMC refined 3D atomic configurations of PMN-35PT. (b) Selected atomic configurations for calculating charge density distribution. The charge density distribution in Pb-O plane (c), and *B*-O plane (d). The strong hybridization between Pb 6*s* with O 2*p* states occurs in every perovskite unit-cell. The Mg-O bonds are ionic type, no hybridization between Mg and O is observed. Covalent bonds of Ti-O, and Nb-O are observed due to the hybridization of their 4*d* with O 2*p* states. These results demonstrate that the local polarization is stabilized mainly by the strong hybridization between Pb 6*s* with O 2*p* states. Besides, the hybridization of Ti/Nb 4*d* with O 2*p* states plays some part.

Comment 3: Fig. 3a It can be clearly seen that the monoclinic structure appears in the disorder phase. The correspondence between 25PT M_B , 30PT M_A , and 35PT M_C is slightly inconsistent between Fig. 3a and Fig 1b. In Fig. 1c, the monoclinic phase is shown to appear at the phase boundary between the rhombohedral and tetragonal phases, thus the importance of the

orthorhombic polarization should be explained. Fig. S2 Volume expansion is necessary to introduce randomness into a mixture of materials consisting of different ion sizes. Is there any indication of this in the average structure obtained by Rietveld analysis?

Reply 3: Thank you for the good comments. Indeed, there are some difference in the polar direction the monoclinic $(110)_p$ plan obtained from the Rietveld analysis against the high-resolution synchrotron X-ray diffraction data and from the RMC fitting. We would like to point that such deviation is reasonable. We note that such deviation has been also observed in PZT⁴. Because these two types of M have the same space group of Cm with the polarization lying on the $\{110\}_p$ mirror plane. The difference is that the polar vector locates between $[001]$ and $[111]$ in M_A and locates between $[110]$ and $[111]$ in M_B . According to the symmetry of crystal structure, the orthorhombic phase should occur. Actually, there is still argument in the existence of long-range O symmetry near MPB of PMN-PT. From the calculated order parameter S of local polar vectors, the S along $[110]$ presents peak value near 30PT-35PT. This leads to the completing state in local polar between R , T , and O . Therefore the local polar order along $[110]$ is very important to form the competing local multi-polar states, and further reduced the anisotropy of free-energy profile.

We agree the review's point that the volume expansion is necessary to introduce randomness. For PMN-PT, the $(Mg^{2+}_{1/3}Nb^{5+}_{2/3})$ has a weighted ionic radius of 0.667 Å, and the ionic radius of Ti^{4+} is 0.605 Å. Therefore, the increasing of $(Mg^{2+}_{1/3}Nb^{5+}_{2/3})$ content will lead to volume expansion, which agree with the increasing local polar disorder as function of composition. According to the Rietveld analysis in Supplementary Fig. 2, the compositional dependence of lattice parameters and the volume of unit cell is shown in Fig. R4. Indeed, the the unit cell volume expands as decreasing Ti content. The volume expansion is necessary to introduce randomness, because more space are needed for cations randomly displacing with respect to their surrounding oxygen.

Fig. R4. (a) Composition-dependent unit cell parameters, (b) unit cell volume of PMN-*x*PT.

Revision: Supplementary Fig. 3 has been added in the Supplementary Information and corresponding description modification has been added in the figure caption. Some discussions are added. Page 9: “Notably, no evidence show the existence of long-range *O* symmetry in the present studied compositions, but the relatively high local polar order occurs along [101]_p direction, which is important to from the competing multi-polar state between [001]_p, [101]_p and [111]_p. Such local competing polar states would be the reason why different long-range structure models were identified near MPBs of PMN-PT, such as the claimed *O* symmetry”

Comment 4: Fig. S7 Pb-L3 XAFS EXAFS oscillations are difficult to observe in PMN-rich XAFS due to structural disorder and phase difference between Mg and Nb. Absorption spectra should be displayed to examine data quality.

Reply 4: We totally agree review’s comment. The high-quality Pb L_{III}-edge EXAFS data are typically difficult to collect. The normalized Pb L_{III}-edge X-ray absorption near edge structure spectra are shown in Fig. S. According to the quality of EXAFS data, the *k*-space range used in the Fourier transform was about 2.1 Å⁻¹ to (8-10) Å⁻¹, while the *r*-space fit was conducted from 1 Å to 3.5 Å. The quality of EXAFS data in present study are comparable with previous reported in the literatures^{5,6}. Actually, during the RMC fitting, the weight factor of Pb L_{III}-edge EXAFS data is smaller than the N-/X-PDF data.

Fig. R5. (a) Normalized Pb L_{III}-edge X-ray absorption near edge structure spectra of PMN-*x*PT. The inset shows the enlarged spectra at absorption edge. (b) The Pb L_{III}-edge XANES spectra of *k*-weighted form of $k^2\chi(k)$ as a function of wavenumber *k* of PMN-*x*PT.

Revision: Supplementary Fig. 8 has been added in the Supplementary Information and corresponding modification has been made.

Comment 5: pS4L64 The minus of “-1/6” in Eq (3) is hard to see.

Reply 5: Thanks for the comment. We have made corresponding modification.

Reviewer #3 (Remarks to the Author):

This manuscript studies the origin of high piezoelectricity in the widely used PMN-PT relaxor ferroelectrics. The authors applied both experimental and simulation tools, including those X-ray scattering, neutron scattering, and phase-field simulations, and concluded that the origin of high piezoelectricity comes from the balanced local polar length and direction randomness. This is an exciting research topic. For a long time, there have been controversial arguments about the major contribution to piezoelectricity in relaxor ferroelectrics. In particular, many models to describe the highest piezoelectricity in MPB have been proposed, but they all stand at different angles with ambiguous conclusions. Electron microscopy has attracted a lot of attention recently to address this issue, but the nature of which focuses on local regions that may not represent the origin as a whole. The current study provides an excellent platform and a new attempt to settle the above-mentioned argument by using various advanced scattering techniques. The manuscript is well-organized and technically sound. I would be more than happy to recommend its publication in Nature Communications upon receiving clarifications from the authors on the following issues.

Reply: We highly appreciate your positive overall comments and publication recommendation.

Comment 1: While both the 30PT and 35PT lie in MPB, their synchrotron X-ray diffraction patterns are completely different. In particular, three curves were demonstrated in Fig. 1c with different peak values and slight peak shifts. To what extent does this represent and what is the reason behind need further clarification.

Reply 1: Thanks for the comments. We would like to point that the super high-resolution synchrotron X-ray diffraction (HRSXRD, $\Delta Q/Q < 1.4 \times 10^{-4}$) in present study enable to detect the peak splitting well in subtle MPB compositions. Conventional Lab XRD (most of previous studies used) is unable to observe these subtle changes. Therefore, both the 30PT and 35PT lie in MPB, the SXRD patterns look obvious different. On the other hand, as shown by the Rietveld refinement against the super high-resolution synchrotron X-ray diffraction data, the 45PT present pure T phase, 35PT present the phase coexistence of M_C and T , 30PT presents the M_A phase. These observations basically agree with pervious results and HRSXRD in literature (Fig. R6). Further, the curves shown in Fig. 1c for 35PT, the circle point indicate the observed data. The red line indicates the fitted data, and the blue and pink peak show the individual peaks of T and M_C , respectively. In order to describe the Fig. 1c clearly, we have revised the corresponding part and modified the figure caption.

Fig. R6. (a) Super high-resolution synchrotron X-ray diffraction patterns of 35PT to 30PT adopted from literature (A. Henriques, Quantifying Crystallographic Structural Uncertainty in Electrically Poled Relaxor Ferroelectrics via Bayesian and Rietveld Refinements, 2022), (b) our work. It can be seen that our results agree well with pervious study. The obviously change in HR-SXRD patterns from 35PT to 30PT is due to composition-sensitive long-range structure changes at MPB.

Comment 2: The nearest neighbour coordination is changed upon changing the Ti content. However, the $G(r)$ of the Ti-O remain almost the same while the position and $G(r)$ of the Nb-O, Mg-O, and PbO pairs are significantly altered with the change of Ti content. The authors need to provide a detailed explanation of this and the related implication.

Reply 2: Thank you for the good comments. Firstly, we would like to give a briefly introduction of the physical significance of peak in PDF. The peak positions give the interatomic spacings in a material. The peak height is proportional to the multiplication of the scattering factors for the two elements. The peak area for one peak, is proportional to the coordination number for that atom-atom pair. Actually, the change of B-O and A-O distance can be seen more clearly from the element-special partial PDFs (as shown Fig. R7(a) for an example), which is obtained from the RMC fitting. Because in the PDF $G(r)$, overlapping exists between Ti-O, Nb-O, and Mg-O peaks. As shown in Figs. R7(b-e), the Pb-O, Ti-O, Mg-O, and Nb-O changes with varying Ti content, but the peak center do not change much. This leads to the mean atomic polar displacement length not change much with composition, but the direction of polar displacement vectors change drastically (Fig. 2b and Fig. 2c). In order to avoid misunderstandings, we have changes the modified the corresponding part and added the Fig. R7 in the supporting information.

Fig. R7. Element specific partial $A/B-O$ PDFs obtained from the refined 3D atomic configuration. (a) Representative partial $A/B-O$ PDFs of 20PT. The compositional dependent partial $A/B-O$ PDFs, (b) Pb-O, (c) Ti-O, (d) Mg-O, (e) Nb-O.

Revision: Supplementary Fig. 10 has been added in the Supplementary Information and corresponding description modifications have been added in the figure caption.

Comment 3: Could the conclusion be extended to other lead-based relaxor ferroelectric systems or even lead-free relaxor ferroelectric systems? I hope the authors can comment on this point.

Reply 3: Thanks for the good comments. We believe the conclusion could be extended to other relaxor ferroelectric systems, including lead-based and lead-free ones based on the following reasons. (1) The lead-based ferroelectric systems, such as $\text{Pb}(\text{Zn}_{1/3}\text{Nb}_{2/3})\text{O}_3\text{-PbTiO}_3$, $\text{Pb}(\text{Sc}_{1/2}\text{Nb}_{1/2})\text{O}_3\text{-PbTiO}_3$, present similar compositional dependent electrical and structural characteristics: 1) they present similar dielectric behavior that the relaxor state evolves into ferroelectric state with increasing PT content; 2) their high piezoelectricity occur at the compositions located at overlapping of long-range MPB and relaxor-ferroelectric boundary, which is a critical macroscopic feature of competing local polar order-disorder between different states. (2) As is widely known, the lead-free relaxors, such as high performance KNN-based piezoelectric systems have similar structure feature with lead-based one, for instance phase boundary and local structure heterogeneity. Therefore, we think our results can be extended to other lead-based and lead-free relaxor ferroelectric systems.

Actually, we are doing some local structure analysis in lead-free relaxors with high piezoelectricity (KNN-based). The preliminary results are shown in Fig. R7, one can see that increasing the local polar disorder (disorder parameters ξ increase from 0.54 in KNN to 0.86 in KNNS) under maintaining the local polar length ($27 \mu\text{C}/\text{cm}^2$ in KNN, and $26 \mu\text{C}/\text{cm}^2$ in KNNS) can largely enhance the piezoelectric d_{33} (d_{33} enhances from 120 pC/N in KNN to ~ 400 pC/N in KNNS relaxor). We have added some discussions in the discussion section.

Fig. R8. The competing local polar order-disorder in KNN-based piezoelectric systems. (a) The disorder polar vector parameter ξ , (b) the local polar length extracted from the 3D atomic configurations, (c) The piezoelectric coefficient d_{33} of KNN and KNNS relaxor ferroelectric. KNN: $\text{K}_{0.5}\text{Na}_{0.5}\text{NbO}_3$; KNNS: $0.96(\text{K}_{0.48}\text{Na}_{0.52})(\text{Nb}_{0.955}\text{Sb}_{0.045})\text{O}_3-0.04(\text{Bi}_{0.5}\text{Na}_{0.5})\text{ZrO}_3$.

Revision: We have added some discussions in the main text.

Page 13: The mechanism proposed in the present work is beyond the prevalent multiphase coexistence mechanism. Further, it is not limited to lead-based relaxor systems, but also can rationalize the current status in lead-free piezoelectric systems, such as KNN-based, BT-based solid solutions.

Page 15: The competing local polar order-disorder mechanism should be applicable to other lead-based and lead-free relaxors, by considering their same physical principles, such as local chemical composition-driven local polar heterogeneity. Further, it is found that the hybridization of Pb $6s$ with O $2p$ states is important to stabilize the large local polar length under a high degree of direction disorder (Supplementary Fig. 17). Therefore, from the perspective of lead-free piezoelectric materials design, introducing the strongly ferroelectric active ions, such as Bi^{3+} , Sb^{5+} , and Sn^{4+} ect., would be beneficial to extend the local polar length. On the other hand, the long-range unit cell volume expansion would be necessary to introduce disorder into a solid solutions consisting of different ion sizes. Thus, it should be considered to increase the local polar disorder.

References

- 1 Wang, S. et al. A phase-field model of relaxor ferroelectrics based on random field theory, *Int. J. Solids Struct.* **83**, 142–153 (2016).
- 2 Hong, Z. et al. Role of point defects in the formation of relaxor ferroelectrics. *Acta Mater.* **225**, 117558 (2022).
- 3 Zhang, Y. et al. New capabilities for enhancement of RMCProfile: instrumental profiles with arbitrary peak shapes for structural refinements using the reverse Monte Carlo method. *J. Appl. Cryst.* **53**, 1509-1518 (2020).
- 4 Zhang, N. et al. Local-scale structures across the morphotropic phase boundary in $\text{PbZr}_{1-x}\text{Ti}_x\text{O}_3$, *IUCrJ* **5**, 73–81 (2018).
- 5 Yoneda, Y. et al. Local structure analysis of relaxor $\text{Pb}(\text{Mg}_{1/3}\text{Nb}_{2/3})\text{O}_3$. *Ferroelectrics* **513**, 1 (2017).
- 6 Eremenko, M. et al. Local atomic order and hierarchical polar nanoregions in a classical relaxor ferroelectric. *Nat. Commun.* **10**, 2728 (2019).

We thank reviewers again for the revision suggestions and all above useful questions and comments. Hopefully the manuscript has been improved by taking into account all above comments and addressing all above questions.

With best regards,

Sincerely yours,

Prof. Dr. Hui Liu on behalf of all authors

Beijing Advanced Innovation Center for Materials Genome Engineering

University of Science and Technology Beijing

Beijing 100083, China.

Tel./Fax: +86-10-62332525

Email: hui.liu@ustb.edu.cn

REVIEWERS' COMMENTS

Reviewer #1 (Remarks to the Author):

The authors have made modifications on improving the manuscript. The responses to comments also were addressed carefully. I think the paper could be accepted.

Reviewer #2 (Remarks to the Author):

Titled "Emergence of high piezoelectricity from competing local polar order-disorder in relaxor ferroelectrics

By Hui Liu, Xiaoming Shi, Yonghao Yao, Huajie Luo, Qiang Li, Houbing Huang, He Qi, Krystian Rolder, Yuanpeng Zhang, Yang Ren, Shelly D. Killy, Jeorg C. Neufeind, Long-Qing Chen, Jun Chen

Appropriate corrections have been made for all points previously pointed out. In addition, there were no contradictory comments on the points pointed out by other referees. There are no further fixes. I would recommend its publication in Nature Communications.

p13 L347 Isn't ect. a mistake of etc.

I examined the validity of the answer to the comment.

However, it is not shown whether the $G(r)$ in Fig. S2 and S4 are resolution-corrected data corresponding to the model assumed in RMC.

Reply: Thank for the good comments. During RMC fitting, the $G(r)$ are considered the resolution effect. The NIST Si SRM powder total scattering data were collected to identify the instrument resolution in both X-ray and neutron total scattering measurements³. The calculated total-scattering data were corrected for the instrument resolution in both reciprocal and real spaces. We have added these missed information in the Supplementary Information, and the figure caption of Supplementary Figs. 6 and 7.

The resolution correction of PDF refinement by RMC modeling is properly mentioned.

Comment 1: XAFS, Neutron PDF, and X-ray PDF are used as experimental data for constructing the local structure model. Of these, strictly speaking, XAFS and N-PDF cannot obtain static real space information. There is a dynamic structural contribution because the energy of the probing source is very close to the vibrational energy of the lattice. The contribution of the dynamic structure to the three kinds of real space information is different, but it is necessary to explain how they were converged to the static structure model.

Reply: Thank for the good comments. We totally agree with the review's point that the contribution of the dynamic structure to the three kinds of real space information is different. Here, we would like to explain the concern by the following:

(i) The reverse Monte Carlo (RMC) routine is a pure data-driven approach, meaning the final model obtained via the combined fitting is a compromise of structural aspects associated with different types of datasets. In another word, the convergence of the model is purely determined by the overall agreement between the calculated patterns and the experimental ones. Specifically, for those datasets containing more significant dynamic contribution (e.g., neutron total scattering, as pointed out by the referee, due to closer energy of thermal neutrons as compared to the low energy lattice vibrations), the corresponding structure solution would contain more dynamic characteristics. On the contrary, the high energy of X-

ray would lead to relatively weaker dynamic contribution in the data set and therefore the correspondingly driven structure model would possess more static characteristics. In practical RMC modeling, the contribution from different datasets (thus the driving force towards different structure characteristics) is controlled by the weight factor and the golden rule, which is what we have been following for our RMC modeling, is that the weight for different datasets should be balanced to guarantee the end result not being biased.

(ii) In the RMC structure modeling, the dynamic and static aspects of the structure is not decoupled – decoupling. The static and dynamic structural aspects is also not the focus of current report and the discussion presented in current report is not based on the distinction between the static and dynamic characteristics of the structure. Furthermore, the discussion presented in current manuscript is not based on the distinction between the dynamic and static factors with respect to the local structural distortion (or, polarization, etc.).

We understand that the structure obtained by RMC modeling is a compromise obtained from N-PDF, X-PDF and XAFS. In addition, since the PDF in the nanometer range discussed in the text has little contribution from the dynamic structure, we determined that similar results could be obtained by incorporating the Placek correction into the N-PDF.

Comment 2: Connecting lattices with different polarizations distorts the lattice and destabilizes the structure. What is the mechanism by which disordered polarization is stabilized? You mention the polarization due to the large Pb displacement due to the hybridization of Pb and oxygen, but can it be extracted from the model structure that Pb inhibits the polarization ordering?

Reply 2: Thank you for the good comments. The local polar heterogeneity is mainly from the local chemical composition fluctuations that different B-site cations (Ti^{4+} , Mg^{2+} , and Nb^{5+}). As widely known, the Pb^{2+} is the strongly ferroelectric active cation. The Ti^{4+} is also strongly ferroelectric active B-site cations, Nb^{5+} is moderately, while Mg^{2+} is weakly ferroelectric active cation. Due to the strong hybridization between Pb 6s with O 2p states, the Pb possesses large polar displacement. The PbTiO_3 (PT) has strong long-range ferroelectric polar ordering (as shown in Fig. R1(a), order parameter S near 1, which corresponds to completely polar order along [001]). While for PMN ($\text{Pb}(\text{Mg}_{1/3}\text{Nb}_{2/3})\text{O}_3$), the long-range polar order is broken, short-range polar order exists. We totally agree that the Pb is important to stabilize the disorder polar. Besides, the ferroelectric active B-site cations also play some role. Because the Pb displacement is affected by the local chemistry. We try to extract the charge density distribution from the model structure, however the size of the model is too big to extract all charge density distribution in the model. We extract the charge density distribution in the part of the structural model. As shown in Supplementary Fig. 17, we can see the strong hybridization between Pb 6s with O 2p states, and inhibiting the polarization ordering. These are added in the Supplementary Fig. 17.

The charge density distribution indicates that not only Pb but also Ti and Nb are hybridized with surrounding oxygen in the structure obtained by RMC modeling. Hybridization with oxygen has been shown to stabilize the distorted structure of perovskite cations other than Mg.

Comment 3: Fig. 3a

It can be clearly seen that the monoclinic structure appears in the disorder orthorhombic polarization should be explained. Fig. S2 Volume expansion is necessary to introduce randomness into a mixture of materials consisting of different ion sizes. Is there any indication of this in the average structure obtained by Rietveld analysis?

Reply 3: Thank you for the good comments. Indeed, there are some difference in the polar direction the monoclinic (110)_p plan obtained from the Rietveld analysis against the high-

resolution synchrotron X-ray diffraction data and from the RMC fitting. We would like to point that such deviation is reasonable. We note that such deviation has been also observed in PZT4. Because these two types of M have the same space group of Cm with the polarization lying on the {110}p mirror plane. The difference is that the polar vector locates between [001] and [111] in MA and locates between [110] and [111] in MB. According to the symmetry of crystal structure, the orthorhombic phase should occur. Actually, there is still argument in the existence of long-range O symmetry near MPB of PMN-PT. From the calculated order parameter S of local polar vectors, the S along [110] presents peak value near 30PT-35PT. This leads to the completing state in local polar between R, T, and O. Therefore the local polar order along [110] is very important to form the competing local multi-polar states, and further reduced the anisotropy of free-energy profile.

We agree the review's point that the volume expansion is necessary to introduce randomness. For PMN-PT, the

($\text{Mg}^{2+}_{1/3}\text{Nb}^{5+}_{2/3}$) has a weighted ionic radius of 0.667 Å, and the ionic radius of Ti^{4+} is 0.605 Å. Therefore, the increasing of ($\text{Mg}^{2+}_{1/3}\text{Nb}^{5+}_{2/3}$) content will lead to volume expansion, which agree with the increasing local polar disorder as function of composition. According to the Rietveld analysis in Supplementary Fig. 2, the compositional dependence of lattice parameters and the volume of unit cell is shown in Fig. R4. Indeed, the unit cell volume expands as decreasing Ti content. The volume expansion is necessary to introduce randomness, because more space are needed for cations randomly displacing with respect to their surrounding oxygen.

We were able to understand the process by which long-distance structures including misalignment due to disorder develop from local structures to monoclinic structures. The above explanation can be understood as a mechanism in which the structure changes even though the structure below the unit cell does not change.

Comment 4: Fig. S7 Pb-L3 XAFS

EXAFS oscillations are difficult to observe in PMN-rich XAFS due to structural disorder and phase difference between Mg and Nb. Absorption spectra should be displayed to examine data quality.

Reply 4: We totally agree review's comment. The high-quality Pb LIII-edge EXAFS data are typically difficult to collect. The normalized Pb LIII-edge X-ray absorption near edge structure spectra are shown in Fig. S. According to the quality of EXAFS data, the k-space range used in the Fourier transform was about 2.1 \AA^{-1} to $(8-10) \text{ \AA}^{-1}$, while the r-space fit was conducted from 1 Å to 3.5 Å. The quality of EXAFS data in present study are comparable with previous reported in the literatures^{5,6}. Actually, during the RMC fitting, the weight factor of Pb L_{III}-edge EXAFS data is smaller than the N-/X-PDF data.

Good evidence of data quality is provided.

Comment 5: pS4L64

The minus of "-1/6" in Eq (3) is hard to see.

Reply 5: Thanks for the comment. We have made corresponding modification.

Confirmed the fix

Reviewer #3 (Remarks to the Author):

My concerns have been well addressed. I will recommend the publication of this work as it is.

Reviewer #1 (Remarks to the Author):

The authors have made modifications on improving the manuscript. The responses to comments also were addressed carefully. I think the paper could be accepted.

Reply: We thank the referee for accepting our paper.

Reviewer #2 (Remarks to the Author):

Appropriate corrections have been made for all points previously pointed out. In addition, there were no contradictory comments on the points pointed out by other referees. There are no further fixes. I would to recommend its publication in Nature Communications.

Reply: We thank the referee for accepting our paper.

Comment: p13 L347 Isn't ect. a mistake of etc.

Reply: Thanks for pointing this mistake. We have corrected it.

Reviewer #3 (Remarks to the Author):

My concerns have been well addressed. I will recommend the publication of this work as it is.

Reply: We thank the referee for accepting our paper.